# Evidential Sparsification of Multimodal Latent Spaces in Conditional Variational Autoencoders

**Masha Itkina, Boris Ivanovic, Ransalu Senanayake, Mykel J. Kochenderfer, Marco Pavone**
Department of Aeronautics and Astronautics
Stanford University
{mitkina, borisi, ransalu, mykel, pavone}@stanford.edu

## Abstract

Discrete latent spaces in variational autoencoders have been shown to effectively capture the data distribution for many real-world problems such as natural language understanding, human intent prediction, and visual scene representation. However, discrete latent spaces need to be sufficiently large to capture the complexities of real-world data, rendering downstream tasks computationally challenging. For instance, performing motion planning in a high-dimensional latent representation of the environment could be intractable. We consider the problem of sparsifying the discrete latent space of a trained conditional variational autoencoder, while preserving its learned multimodality. As a post hoc latent space reduction technique, we use *evidential theory* to identify the latent classes that receive direct evidence from a particular input condition and filter out those that do not. Experiments on diverse tasks, such as image generation and human behavior prediction, demonstrate the effectiveness of our proposed technique at reducing the discrete latent sample space size of a model while maintaining its learned multimodality.

## 1  Introduction

Variational autoencoders (VAEs) with discrete latent spaces have recently shown great success in real-world applications, such as natural language processing [1], image generation [2, 3], and human intent prediction [4]. Discrete latent spaces naturally lend themselves to the representation of discrete concepts such as words, semantic objects in images, and human behaviors. The choice of a discrete latent space encoding over a continuous one has also been shown to encourage multimodal predictions [5, 6] as well as interpretability [7, 8] since it is easier to analyze input-output relationships on countable classes than continuous vector spaces [9]. However, prohibitively large discrete latent spaces are required to accurately learn complex data distributions [10, 11], thereby causing difficulties in interpretability and rendering downstream tasks computationally challenging. For instance, robotic motion planning algorithms often plan using a uniform distribution over the state space representation, requiring exorbitantly many samples to accurately cover a large latent space [12]. Similarly, in multi-agent robotics, a high dimensional latent space encoding may be too large to transmit over limited bandwidth between coordinating robots [13]. In an attempt to address these concerns, we propose a methodology that effectively reduces the latent sample space size while maintaining multimodality within the latent distribution.

Distributional multimodality arises in many real-world problems, such as video frame prediction [14] and human behavior prediction [8, 15], from multiple possibilities for the future (e.g., a pedestrian may turn right or left given the same trajectory history). The conditional variational autoencoder (CVAE) was developed to address prediction multimodality [16]. The CVAE encodes input data $x$ and corresponding query label $y$ into a latent space $Z$. At test time, the model samples from the latent space encoding of a query label to generate diverse data. The latent space distribution $p(z \mid y)$

encodes the multimodality of the prediction task. During inference, the discrete latent distribution in a CVAE is often parameterized by the softmax function. A drawback to the softmax transformation is that uncertainty is distributed across all the available classes since, by definition, the softmax function cannot set a probability to zero (though it can become negligible). Removing latent classes that arise within the distribution solely due to this uncertainty has the potential to reduce the number of relevant latent classes to consider.

**Contributions**   We introduce a novel method grounded in evidential theory for sparsifying the discrete latent space of a trained CVAE. Evidential theory, also known as Dempster-Shafer Theory (DST), differentiates lack of information (e.g., an uninformative prior) from conflicting information (e.g., evidence supporting multiple hypotheses) by considering the power set of hypotheses [17]. We propose using evidential conflict as a proxy for multimodality. We can then prune the latent classes from the distribution that do not directly receive evidence from the input features, thus performing post hoc latent space sparsification without sacrificing distributional multimodality. Experiments show that our algorithm achieves a significant reduction in the discrete latent sample space of CVAE networks trained for the tasks of image generation and behavior prediction, without loss of network performance. Our approach sparsifies the latent space while maintaining distributional multimodality, unlike baseline techniques which remove important modes with overly aggressive filtering. Our proposed method provides a more accurate distribution over the latent encoding with fewer training iterations than baseline methods, and demonstrates consistent performance on downstream tasks.

## 2   Evidential Theory for Latent Space Reduction

Evidential theory distinguishes lack of information from conflicting information [17], making it appealing for handling epistemic uncertainty in machine learning tasks [18]. Denoeux [19] recently showed that, under a set of assumptions, the softmax transformation is equivalent to the Dempster-Shafer fusion of belief masses. This insight facilitates the use of evidential theory in multi-class machine learning classification paradigms. We propose using the tools from evidential theory to sparsify discrete latent spaces in CVAEs. Our method automatically balances the objectives of sparsity and multimodality by keeping only the latent classes that receive direct evidence from the network's features and weights. The following sections overview evidential theory, its application to neural network classifiers, and our proposed approach to evidential latent space sparsification in CVAEs. Further information on evidential theory can be found in Appendix A.

### 2.1   Evidential Theory

**Mass Functions**   Evidential theory considers a discrete set of hypotheses or, equivalently, classes. Let the set of allowable classes be $Z = \{z_1, \ldots, z_K\}$, where $z_k$ can be represented as one-hot encodings, and denote its power set by $2^Z$. A belief mass is a function $m : 2^Z \to [0, 1]$ such that $\sum_{A \subseteq Z} m(A) = 1$ [17]. Evidential theory assumes that the allowable classes are exhaustive, that is $m(\emptyset) = 0$ [20]. The mass function quantifies the total evidential belief committed to some $A \subseteq Z$. If $m(Z) = 1$, then the mass function is vacuous, in that it encodes a complete lack of evidence for any particular subset of classes. If the belief mass function is non-zero only for singleton sets, then it reduces to an approximation of the usual categorical distribution. Two mass functions, each representing an independent source of evidence, can be combined through Dempster's rule [17] to generate a fused belief mass as defined in Appendix A.3.

A belief mass function $m$ is *simple* if there is at most a single strict subset $A \subset Z$ for which $m(A)$ is non-zero. Then, we can define,

$$m(A) = s, \quad m(Z) = 1 - s, \quad w = -\log(1 - s), \tag{1}$$

where $s \in [0, 1]$ is the degree of support in $A$ and $w$ is the corresponding *evidential weight* for $A$ [20].

**Plausibility Transformation**   Plausibility $pl(z_k)$ represents the extent to which the evidence does not contradict the class $z_k$ [21]. Belief masses can be reduced to estimated probabilities through the plausibility transformation [22],

$$p(z_k) = \frac{pl(z_k)}{\sum_{l=1}^{K} pl(z_l)}, \tag{2}$$

where $pl(z_k) = \sum_{B:z_k \in B, B \subseteq Z} m(B)$ and $k \in \{1, \ldots, K\}$.

## 2.2 Evidential Classifiers

It has been shown that all classifiers that transform a linear combination of features through the softmax function can be formulated as evidential classifiers [19]. Each feature represents an elementary piece of evidence in support of a class or its complement. The softmax function then fuses these pieces of evidence to form class probabilities conditioned on the input. In this context, the softmax class probabilities are equivalent to normalized plausibilities (Eq. (2)). Thus, the neural network weights and features, which serve as arguments to the softmax function, can also be used to compute the corresponding belief mass function. When compared to a Bayesian probability distribution, a belief mass function provides an additional degree of freedom that allows it to distinguish between a lack of evidence and conflicting evidence.

A feature vector $\phi(y_i) \in \mathbb{R}^J$ is defined as the output of the last hidden layer in a neural network, for a given query $y_i$ from a dataset. The evidential weights defined in Eq. (1) are assumed to be affine transformations of each feature $\phi_j(y_i)$ by construction,

$$w_{jk} = \beta_{jk}\phi_j(y_i) + \alpha_{jk}, \tag{3}$$

where $\alpha_{jk}$ and $\beta_{jk}$ are parameters [19]. An assumption is made that the evidence supports at most either a singleton class $\{z_k\}$ when $w_{jk}^+ = \max(0, w_{jk}) > 0$ or its complement $\overline{\{z_k\}}$ when $w_{jk}^- = \max(0, -w_{jk}) > 0$, such that $w_{jk}^+ - w_{jk}^- = w_{jk}$. Then, for each feature $\phi_j(y_i)$ and each class $z_k$, according to Eq. (1), there exist two simple mass functions,

$$m_{kj}^+(\{z_k\}) = 1 - e^{-w_{jk}^+}, \quad m_{kj}^+(Z) = e^{-w_{jk}^+} \tag{4}$$

$$m_{kj}^-(\overline{\{z_k\}}) = 1 - e^{-w_{jk}^-}, \quad m_{kj}^-(Z) = e^{-w_{jk}^-}. \tag{5}$$

These masses can then be fused through Dempster's rule to arrive at the mass function at the output of the softmax layer as follows,

$$m(\{z_k\}) = Ce^{-w_k^-}\left(e^{w_k^+} - 1 + \prod_{\ell \neq k}\left(1 - e^{-w_\ell^-}\right)\right) \tag{6a}$$

$$m(A) = C\left(\prod_{z_k \notin A}\left(1 - e^{-w_k^-}\right)\right)\left(\prod_{z_k \in A}e^{-w_k^-}\right), \tag{6b}$$

where $A \subseteq Z, |A| > 1$, $C$ is a normalization constant, $w_k^- = \sum_{j=1}^J w_{jk}^-$, and $w_k^+ = \sum_{j=1}^J w_{jk}^+$.

## 2.3 Evidential Sparsification

Under the assumptions outlined in Section 2.2 and using the plausibility transformation, the equivalence of the mass function in Eq. (6) and the softmax distribution holds under the constraint $\sum_{j=1}^J \alpha_{jk} = \hat{\beta}_{0k} + c_0$, where $\hat{\beta}_{0k}$ are the bias parameters learned by the neural network and $c_0$ is a constant [19]. The evidential weight parameters $\alpha_{jk}$ and $\beta_{jk}$ in Eq. (3) are not uniquely defined due to the extra degree of freedom provided by the belief mass as compared to the softmax distribution. Denoeux [19] selects the $\alpha_{jk}$ and $\beta_{jk}$ parameters that maximize the Least Commitment Principle (LCP), which is analogous to maximum entropy in information theory [23].

In contrast, we choose our parameters such that the singleton mass function in Eq. (6a) is sparse, rather than distributing uncertainty across the mass function using the LCP. We construct the singleton mass function to identify only the classes that receive direct evidence towards them. We observe that if $w_k^+ = 0$ and $w_\ell^- = 0$ for at least one other class $\ell \neq k$, then $m(\{z_k\}) = 0$. Intuitively, if no evidence directly supports class $k$ and there is no evidence contradicting another class $\ell$, then the belief mass for the singleton set $\{z_k\}$ is zero. This situation occurs when at least one of $w_k^+$ and $w_k^-$ is zero, which holds if $w_k^+ = \max(0, w_k)$ and $w_k^- = \max(0, -w_k)$, where $w_k = \sum_{j=1}^J w_{jk}$. Hence, $w_k$ provides direct support either for or against a class $k$. This property does not hold in the original formulation by Denoeux [19]. Thus, we construct an evidential weight $w_{jk}$ that does not depend on $j$, enforcing this desideratum:

$$w_{jk} = \frac{1}{J}\left(\beta_{0k} + \sum_{j=1}^J \beta_{jk}\phi_j(y_i)\right). \tag{7}$$

Since $w_{jk}$ is constant across the index $j$, summing over $j$ in $w_{jk}^+$ and $w_{jk}^-$ yields $w_k^+ = \max(0, w_k)$ and $w_k^- = \max(0, -w_k)$, as required. The corresponding parameters in Eq. (3) are then:

$$\beta_{jk} = \hat{\beta}_{jk} - \frac{1}{K} \sum_{l=1}^{K} \hat{\beta}_{j\ell} \tag{8}$$

$$\alpha_{jk}(y_i) = \frac{1}{J} \left( \beta_{0k} + \sum_{j=1}^{J} \beta_{jk} \phi_j(y_i) \right) - \beta_{jk} \phi_j(y_i), \tag{9}$$

where $\hat{\beta}_{jk}$ are the output linear layer weights learned by the neural network a priori. These parameters match those of Denoeux [19], except $\phi_j(y_i)$ replaces $\mu_j = \frac{1}{N} \sum_{i=1}^{N} \phi_j(y_i)$ in $\alpha_{jk}$. The $\alpha_{jk}$ bias term is now a function of the test input query $y_i$. By choosing to treat each input $y_i$ individually at test time, we remove the dependency in $w_{jk}$ on $j$, facilitating our desired behavior in the singleton mass function. We show that the new $\alpha_{jk}$ and $\beta_{jk}$ parameters satisfy the constraint to achieve equivalency with the softmax transformation in Appendix B. We posit that filtering out the classes with zero singleton belief mass values according to the proposed definition removes only the classes without direct evidence in their support, while imposing a more concentrated distribution output.

## 2.4 Post Hoc CVAE Latent Space Sparsification

Since the discrete latent distribution in a CVAE is parameterized using a softmax function at test time, we can directly apply the evidential theory formulation developed in Sections 2.1–2.3 to sparsify the latent space of a trained CVAE. The evidence allocated to multiple singleton latent classes indicates *internal conflict* between them within the mass function. In the context of a CVAE, we posit that internal conflict is directly correlated with latent space multimodality. High evidential conflict between a subset of latent classes indicates distinct, multimodal latent features encoded by the network. We propose filtering the latent distribution to maintain only these highly conflicting classes, thus, reducing latent sample space dimensionality, without compromising the captured multimodality.

Using Eq. (6a) and Eq. (7), we construct the singleton mass function that corresponds to the encoder's output softmax distribution $p(z \mid y)$ over the latent classes $z_k$ given an input query $y$. This distribution is filtered by removing the probabilities for the latent classes with zero singleton mass values and then renormalizing, as follows,

$$p_{\text{filtered}}(z_k \mid y) = \frac{\mathbb{1}\{m(\{z_k\}) \neq 0\} p_{\text{softmax}}(z_k \mid y)}{\sum_{\ell=1}^{K} \mathbb{1}\{m(\{z_\ell\}) \neq 0\} p_{\text{softmax}}(z_\ell \mid y)}. \tag{10}$$

In this manner, we reduce the number of relevant latent classes, providing a more concentrated latent class distribution, while maintaining the learned distributional multimodality.

# 3 Experiments

To validate our method, we consider CVAE architectures designed for the tasks of class-conditioned image generation and pedestrian trajectory prediction. These real-world tasks require modeling high degrees of distributional multimodality. We compare our method to the softmax distribution and the popular class-reduction technique termed *sparsemax* which achieves a sparse distribution by projecting an input vector onto the probability simplex [24]. By design, both our method and sparsemax compute an implicit threshold for each input query post hoc. Thus, they do not need to be tuned for each network or dataset, and automatically adapt to individual input features. Experiments indicate that our method is able to better balance the objectives of sparsity and multimodality than sparsemax by keeping only the latent classes that receive direct evidence from the network's features and weights, as described in Section 2. We demonstrate that our method maintains distributional multimodality, unlike sparsemax, whilst yielding a significantly reduced latent sample space size over softmax. The code to reproduce our results can be found at: https://github.com/sisl/EvidentialSparsification.

## 3.1 Image Generation

To gain insight into our proposed approach, we run experiments on a small network trained on MNIST [25]. We then demonstrate the performance of our sparsification algorithm on the large

discrete latent space within the state-of-the-art VQ-VAE [2] architecture trained on *mini*ImageNet [26]. All image generation experiments were run on a single NVIDIA GeForce GTX 1070 GPU.

### 3.1.1 MNIST Experiments

**Task** To investigate discrete, multimodal latent representations, we consider the multimodal task of generating digit images for *even* and *odd* queries ($y \in \{$even, odd$\}$) on MNIST. Appendix F and Appendix G contain results on Fashion MNIST [27] and NotMNIST [28], respectively.

**Model** We present a proof of concept of our method for sparsifying multimodal discrete latent spaces on the CVAE architecture shown in Fig. 1. We intentionally use a simplistic CVAE architecture trained on a reasonably simple task to 1) demonstrate the capability of our latent space reduction technique to improve performance post hoc and 2) easily characterize the results.

During training, the encoder consists of two multi-layer perceptrons (MLPs). One MLP takes as input the query $y$, and outputs a softmax distribution that parameterizes the categorical prior distribution $p(z \mid y)$ over the latent variable $z$. The other MLP takes as input a feature vector $x$ and the query $y$, and outputs the softmax distribution for the posterior $q(z \mid x, y)$. The latent class is sampled from $q$ during training and $p$ at test time. It is then passed through the decoder MLP to generate the feature vector $x'$. The Gumbel-Softmax distribution is used to backprop-agate gradients through the discrete latent space [7, 29]. The model is trained to maximize the standard conditional evidence lower bound (ELBO) [16]. Further experimental details are provided in Appendix C.

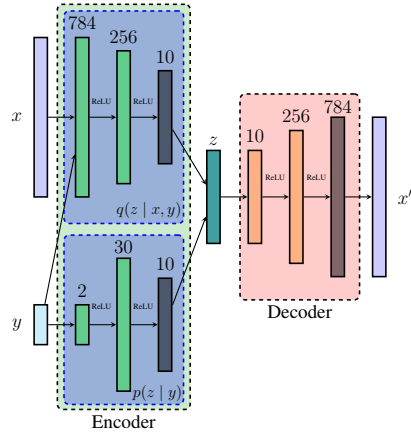

Figure 1: The CVAE architecture used for MNIST image generation. The last layer in each MLP is a softmax layer. At test time, $p(z \mid y)$ is used to sample the latent space.

**Qualitative Performance** We choose $K = 10$ latent classes; with a "perfectly trained" network, this would yield a 5-modal distribution when conditioned on one of $y \in \{even, odd\}$. For instance, conditioning on the *even* query should produce a uniform distribution over the encoded digits: 0, 2, 4, 6, and 8. Fig. 2 depicts the latent distributions of our proposed method, softmax, and sparsemax for the *even* and *odd* queries. Although the CVAE in Fig. 1 successfully learns a multimodal latent encoding, the learned softmax distribution has non-negligible probability mass associated with the incorrect latent classes $z_k$ for each query class $y$. Thus, sampling from the CVAE with the softmax distribution results in an imperfect set of generated digit images given a query as shown in Appendix D.

To remedy this problem, we consider both sparsemax and our proposed evidential filtration technique. In Fig. 2, our filtered distribution selects an almost perfect set of correct latent classes given a query. The proposed distribution provides a more robust signal to the decoder network, improving the accuracy of the sampled images as demonstrated in Appendix D. Thus, we successfully sparsify the

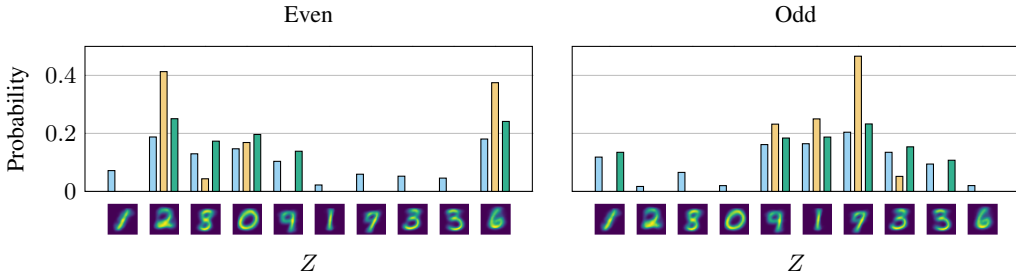

Figure 2: Our proposed filtered distribution (green) yields a more accurate distribution on the MNIST dataset than softmax (blue) and sparsemax (orange). Our method reduces the size of the relevant latent sample space without removing valid latent classes. The horizontal axis depicts the decoded image for each latent class.

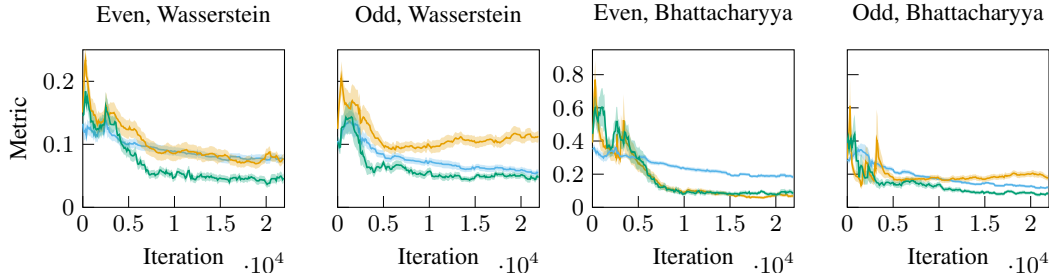

Figure 3: Our filtered distribution (green) outperforms the softmax (blue) and sparsemax (orange) baselines across training iterations on the MNIST dataset. Lower is better.

latent space analytically for each query without knowledge of or comparison to the other query. The only error made by the filtered distribution is the selection of the 9 image (fifth latent class) for the *even* query. This error can be explained by the relatively high softmax probability assigned to the 9 image for both the *even* and *odd* queries. The key insight to our approach is that it performs only as well as the quality of the representation learned by the neural network, with the benefit of extracting richer information than softmax. In contrast, sparsemax results in undesirably more aggressive filtering than our method. It removes the correct latent classes of 1 and 3 for the *odd* query. Both our method and sparsemax result in a filtration decision through an implicit thresholding of the softmax distribution internal to each individual method. Our filtration technique based in evidential theory results in an empirically lower implicit threshold than that of sparsemax. Thus, our more conservative filtration is a compelling latent space reduction technique, particularly for applications where false negatives might cause safety concerns. One such application is human behavior prediction in the context of autonomous driving, which we consider in Section 3.2.

As a further thought experiment, we consider a static threshold that could be chosen through hyperparameter tuning on a validation set. By visual inspection, it is impossible to choose a single static threshold that would outperform our method in balancing sparsity and multimodality in Fig. 2 (either additional false negatives or false positives would result). While we could tune a static threshold for each individual input query for the MNIST task, this would be intractable for a continuous trajectory input query as in the behavior prediction task in Section 3.2.

**Quantitative Evaluation Metrics** Quantitatively evaluating the performance of the filtered distribution is nontrivial as the ground truth distribution can be ambiguous (e.g., a generated image may look like a 3 or an 8). To standardize the evaluation for this binary-input task, we introduce the following metric. The target probability for an input is set to zero if the learned softmax probability conditioned on the input $p(z_k \mid y)$ is smaller than its complement $p(z_k \mid \bar{y})$. If it is larger, then the target probability takes the value of $p(z_k \mid y)$. The result is then normalized across latent classes. Thus, we obtain a sparse, multimodal distribution over the more prominent classes as learned by the network. We write the target distribution as $p_T(z_k \mid y) = \frac{\mathbb{1}\{p(z_k|y) \geq p(z_k|\bar{y})\}p(z_k|y)}{\sum_{\ell=1}^{K} \mathbb{1}\{p(z_\ell|y) \geq p(z_\ell|\bar{y})\}p(z_\ell|y)}$. We use the Wasserstein and Bhattacharyya distances to the target distribution as evaluation metrics. The Kullback-Leibler divergence is not used as it is undefined for zero probabilities.

**Training Iteration Evolution** Fig. 3 demonstrates the robustness of the filtered distribution to fewer training iterations in its ability to extract more accurate encoding information from the neural network earlier in the training process than softmax[1]. Our methodology provides significant performance improvement over the softmax baseline when the latter assigns non-negligible probability mass to incorrect latent classes given a query, as is the case for the *even* query as shown in Fig. 2 and Fig. 3. Although the performances of sparsemax and our proposed distributions are similar for the *even* query on the Bhattacharyya metric, sparsemax significantly underperforms for the *odd* query, even as compared to softmax. The aggressive filtration within sparsemax incorrectly filters out potentially valid latent classes for the *odd* query. Thus, our method consistently outperforms both baselines on MNIST, on average resulting in a $22\%$ improvement over softmax and a $10\%$ improvement over sparsemax on the Bhattacharyya metric. Thus, the proposed latent distribution provides a more robust

Table 1: Downstream classification performance on 1600 sampled images (25 samples × 64 classes) shows that our sparse distribution maintains the original softmax performance, unlike sparsemax. For comparison, the classifier was evaluated on a held-out subset of 1920 images from the original *mini*ImageNet training set. Higher is better for all metrics and bold highlights the best performing latent distributions.

|  | **Softmax** | **Sparsemax** | **Ours** | **Original Images** |
|---|---|---|---|---|
| Accuracy (%) | **20.688** | 6.125 | **19.937** | 71.719 |
| Top 5 Class Accuracy (%) | **47.750** | 17.500 | **47.875** | 90.625 |

representation, retrieving richer information from the learned neural network weights with fewer training iterations. We present further experiments on a reduced data model in Appendix E.

### 3.1.2 VQ-VAE Experiments

We demonstrate the performance of our sparsification methodology on a much larger latent sample space within the state-of-the-art VQ-VAE [2] image generation architecture. The VQ-VAE architecture is trained in two stages. First, the encoder-decoder is trained assuming a uniform prior over the discrete latent space. Then an autoregressive prior is trained over the latent space to allow for sampling from the network. We use *mini*ImageNet images randomly cropped to $128 \times 128$ pixels for training as opposed to ImageNet [30] due to limited computational resources. We consider a latent space of $32 \times 32$ discrete latent variables with $K = 512$ classes each. As in the original paper [2], we train a PixelCNN [31] network for the prior, but reduce its capacity to 20 layers with a hidden dimension size of 128. Further experimental details can be found in Appendix H[2].

Our algorithm achieves an 89% reduction in the 512 latent classes required to represent each latent variable, while maintaining the multimodality of the distribution. On the other hand, sparsemax achieves a 99% reduction in the latent sample space, but sacrifices multimodality, resulting in degenerate single color images sampled from the autoregressive prior. Although sparsemax reduces the latent sample space by a larger percentage than our technique, this negatively impacts its performance due to undesirable pruning of correct latent classes. To further evaluate the performance of our proposed sparse latent distribution, we consider the downstream task of image classification. We train a Wide Residual Network (WRN) [32] for classification on the *mini*ImageNet data. We generate a dataset of 25 examples from each of the 64 training classes in *mini*ImageNet by sampling from the prior. The decoded images are then classified by the WRN. Table 1 shows that our much smaller latent space successfully maintains the performance of the original softmax distribution.

### 3.2 Behavior Prediction

We show that our sparse latent sample space allows for easier interpretability and maintains distributional multimodality on the difficult task of pedestrian trajectory prediction. Trajectron++ [4] is a state-of-the-art deep probabilistic generative model of pedestrian trajectories. It is a graph-structured recurrent model that produces a distribution over future trajectories given an agent's past trajectory history and the past trajectories of its neighboring agents. The model uses a CVAE to capture the multimodality over future trajectory predictions, with a latent space comprised of two discrete variables with five classes each, resulting in 25 total latent classes. The loss function is comprised of the classic conditional ELBO loss in a $\beta$-VAE [33] scheme and a mutual information loss term on $z$ and $y$ as per [34]. We evaluate Trajectron++'s performance with different probability filtering schemes on 203 randomly-sampled examples from the test set of the ETH pedestrian dataset [35], consisting of real world human trajectories with rich interaction scenarios. Behavior prediction model training and experiments were performed on two NVIDIA GTX 1080 Ti GPUs. Further experimental details are provided in Appendix I.

We demonstrate our method's performance as compared to the softmax and sparsemax baselines in Fig. 4. Our filtered latent space kept $2 - 12$ latent classes out of 25 total (51.7% of the test set resulted in 6 maintained latent classes), achieving more than a 50% reduction. Despite the significant sample space reduction, our filtered distribution successfully captures the ground truth when the learned latent space encompasses it, while maintaining the multimodality of the output as seen in Fig. 4. For instance, in Fig. 4a and Fig. 4c, our method identifies two distinct modes where the pedestrian is

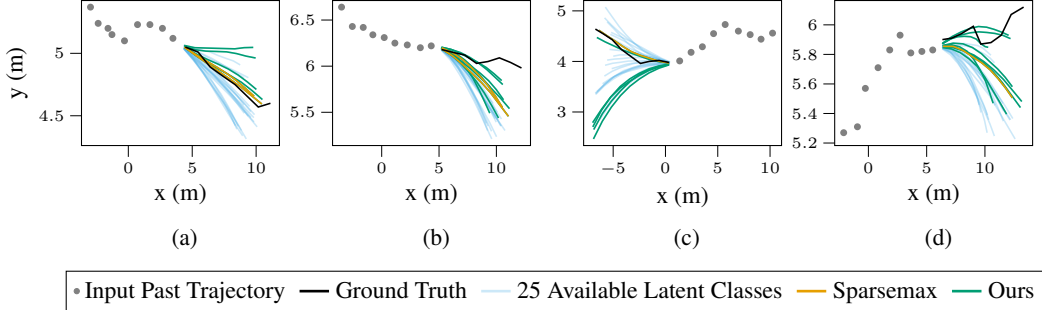

Figure 4: Behavior prediction results on the ETH pedestrian dataset [35] show that our method selects distinct, interpretable modes in the latent space while capturing the ground truth. In contrast, sparsemax occasionally misses the ground truth due to its aggressive filtering scheme. We averaged 100 samples from the Trajectron++'s output for each latent class.

predicted to follow an approximately straight path or choose to turn, both visually valid options. The reduced number of trajectories and the appearance of distinct modes in the filtered output aids with the interpretability of the latent space.

We consider two quantitative experiments in Table 2: 1) sampling from the network according to each of the softmax, sparsemax and our latent distributions and 2) considering the five most likely latent classes according to each distribution and taking the best metric across them. We use standard trajectory prediction metrics to evaluate performance: the Average Displacement Error (ADE), Final Displacement Error (FDE), and Negative Log Likelihood (NLL) [4]. In the sampling experiment, both our proposed distribution and sparsemax quantitatively perform similarly on average to the original softmax distribution, but with the benefit of a significantly reduced latent sample space. However, sampling may not cover a sufficient number of modes due to high likelihoods in a single class. In the second experiment, our method maintains the same performance as softmax, but outperforms sparsemax due to the latter generating false negatives, and collapsing the captured multimodality. As seen qualitatively, although, sparsemax results in a sparser latent distribution, it filters out potentially valid modes as in Fig. 4b and Fig. 4d, reducing its applicability in such safety critical applications as behavior prediction in the context of autonomous driving.

## 4  Related Work

In recent years, a number of new perspectives on the softmax function have been presented. The Gumbel-Softmax distribution was introduced to allow backpropagation through categorical distributions, giving rise to the popularity of discrete latent spaces within CVAE architectures [7, 29]. Related works in low-dimensional encodings for VAEs generally focus on regularization [36] and enforcing structure in the latent space [37] during training, but they do not sparsify the latent space post hoc. Sensoy et al. [38] present an evidential approach to epistemic uncertainty estimation in neural networks for classification tasks. They propose learning Dirichlet distribution parameters to form a distribution over softmax functions. The Dirichlet parameters serve as evidence towards singleton classes, resulting in a loss that regularizes misleading evidence towards the vacuous mass. Unlike Sensoy et al. [38], we focus on post hoc distributional sparsification rather than capturing epistemic uncertainty. Duch and Itert [39] suggest a post hoc modification to further disperse uncertainty among the classes, thus *flattening* the softmax distribution to improve classification performance. Conversely, we are interested in removing the classes that have probability mass assigned to them due to uncertainty rather than evidence, in this way generating a more *sharply peaked* distribution.

Martins and Astudillo [24] introduce the sparsemax distribution, which follows similar motivations to our work in 1) filtering large output spaces and 2) addressing multimodality in classification tasks. An important distinction between our work and sparsemax is the formulation of the problem to arrive at the sparse distribution. Sparsemax finds the Euclidean projection of the input onto the probability simplex. In contrast, we identify a sparse distribution by means of evidential theory, filtering the classes that do not have direct evidence towards them as determined by the learned weights of the neural network. Empirically, we demonstrate that sparsemax results in undesirably more aggressive filtering than our method, making it less effective for applications where false negatives could result

Table 2: Our proposed distribution maintains performance with softmax, while sparsifying the latent space. It preserves the multimodality of the original distribution, unlike sparsemax, which collapses the multimodality, as seen when computing the minimum over the top five most likely latent modes. Direct sampling metrics were computed over 2000 samples from the Trajectron++ network and the top five metrics were computed over 500 samples per latent class. For all metrics lower is better and the best performance is highlighted in bold.

|  | Softmax | Sparsemax | Ours |
|---|---|---|---|
|  | | Direct Sampling | |
| NLL | $4.698 \pm 0.443$ | $4.698 \pm 0.443$ | $4.686 \pm 0.453$ |
| ADE | $0.558 \pm 0.001$ | $0.558 \pm 0.001$ | $0.559 \pm 0.001$ |
| FDE | $1.141 \pm 0.002$ | $1.141 \pm 0.002$ | $1.142 \pm 0.002$ |
|  | | Top 5 Sampling | |
| NLL | $\mathbf{3.951 \pm 0.425}$ | $4.360 \pm 0.422$ | $\mathbf{3.862 \pm 0.419}$ |
| ADE | $\mathbf{0.376 \pm 0.021}$ | $0.397 \pm 0.021$ | $\mathbf{0.376 \pm 0.021}$ |
| FDE | $\mathbf{0.757 \pm 0.049}$ | $0.802 \pm 0.051$ | $\mathbf{0.753 \pm 0.050}$ |

in a reduction of safety. Laha et al. [40] augment the sparsemax technique to control sparsity, presenting sparsegen-lin (sparsemax with regularization) and sparsegen-hg (sparsemax with a scaled input). Both of these methods require hyperparameter tuning to achieve a sparse distribution post hoc which may be computationally prohibitive on larger networks. Our method is able to automatically balance the objectives of sparsity and multimodality while adapting to different feature inputs without hyperparameter tuning. Concurrently to our work, Correia et al. [41] introduced a sparsemax-based method for backpropagating through a discrete latent space as an alternative to the Gumbel-Softmax relaxation. Correia et al. [41] focus on latent space sparsification to allow for efficient marginalization during training, while our work focuses on post hoc latent space sparsification that preserves the learned multimodality.

## 5   Conclusions

In this work, we present a fully analytical methodology for post hoc discrete latent space sparsification in CVAEs. The proposed filtered distribution outperforms the ubiquitous softmax and sparsemax distributions in experiments, extracting more accurate information with fewer training iterations, while improving interpretability and significantly reducing the latent sample space size during inference. We leave the investigation of evidential latent space reduction at training time to future work.

## Broader Impact

Our work focuses on sparsifying the latent space of a conditional variational autoencoder (CVAE). We consider the tasks of image generation and pedestrian trajectory prediction for empirical evaluation. We intend our work to be applicable in the domain of robotics. For instance, our method has the potential to improve tractability of motion planning in a latent representation of the robot's dynamics and to decrease the amount of information required to be transmitted between coordinating robots. However, our work is also more broadly applicable to any domain that would benefit from sparsifying the discrete latent distribution of a pre-trained CVAE. As such, in the process of sparsification, any inherent bias in the pre-trained network may be amplified, causing potential ethical concerns. Also, although we extensively validate our work empirically, we do not provide theoretical safety guarantees for the removed latent classes, requiring sufficient safety testing for any downstream task. We hope that our contribution will enable future positive research outcomes within the fields of robotics, generative modeling, and evidential theory.

## Acknowledgments and Disclosure of Funding

We thank Tim Salzmann for helping us with the behavior prediction experiments. We thank Prof. Katherine Driggs-Campbell and Spencer M. Richards for their valuable feedback. We thank Dr. Boris Kirshtein for his advice and assistance. Toyota Research Institute ("TRI") provided funds to assist the authors with their research but this article solely reflects the opinions and conclusions of its authors and not TRI or any other Toyota entity.

## Footnotes

[1]The results are over 25 different random seeds.

[2]We largely follow the training procedure here: `https://github.com/ritheshkumar95/pytorch-vqvae/`.

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
