[Supplementary Material · NeurIPS_2020_Evidential_CVAEs_Final_Draft_01082020_Appendix.pdf]

# A   Evidential Theory

## A.1   Supplementary Background

Evidential theory diverges from Bayesian probability theory by modifying Kolmogorov's definition of a probability measure [20], as follows:

**Definition A.1.** Given a finite set $Z = \{z_1, \ldots, z_K\}$ and the power set $2^Z$, the evidential belief function $Bel : 2^Z \to [0, 1]$ satisfies the following conditions:

1. $Bel(\emptyset) = 0$

2. $Bel(Z) = 1$

3. For every positive integer n and every collection $A_1, \ldots, A_n$ of subsets of $Z$,

$$Bel(A_1 \cup An) \geq \sum_i Bel(A_i) - \sum_{i<j} Bel(A_i \cap A_j + \ldots \tag{11}$$
$$+ (-1)^{n+1} Bel(A_1 \cap \ldots \cap A_n).$$

The conditions for an evidential belief function are identical to those in Bayesian theory, with the exception that the third condition is relaxed to a lower bound rather than an equality [20]. The evidential belief assigned to a set $A \subseteq Z$ as defined above includes the belief committed to any subset of $A$, as well. If we want to consider the belief assigned to exactly the set $A$, we use the concept of a basic probability assignment or a belief mass function [20], as follows:

**Definition A.2.** Given a finite set $Z = \{z_1, \ldots, z_K\}$ and the power set $2^Z$, the basic probability assignment or evidential belief mass function $m : 2^Z \to [0, 1]$ satisfies the following conditions:

1. $m(\emptyset) = 0$

2. $\sum_{A \subseteq Z} m(A) = 1$.

The belief function, which is also referred to as a *lower probability* of $A$, can also be expressed in terms of the mass function, as follows:

$$Bel(A) = \sum_{B \subseteq Z} m(B). \tag{12}$$

In parallel, the *upper probability* of $A$, or the plausibility of A [20], is defined as follows:

$$Pl(A) = 1 - Bel(\overline{A}). \tag{13}$$

The existence of upper and lower probabilities differentiates evidential theory from Bayesian methods. Evidential theory is able to distinguish between lack of evidence towards a hypothesis and evidence against a hypothesis. Thus, the belief function indicates the total belief committed to the set $A$ and its subsets [18], whereas plausibility is the amount of evidence *not* against $A$ [18].

## A.2   Illustrative Example

To provide better intuition for evidential theory, we outline an example here. Consider a region in space that may be occupied by an obstacle. Let $z_1$ correspond to the hypothesis that the region is *occupied*, $z_2$ to the hypothesis that the region is *free*, and $Z = \{z_1, z_2\}$. Then, we have $2^Z = \{\emptyset, \{z_1\}, \{z_2\}, Z\}$. The belief mass assigned to the set $Z$ is an indication of uncertainty or lack of evidence. Thus, when there are no sensor measurements, we do not have any information, and can assign the entirety of the mass to the unknown set $Z$. Suppose, we then receive many conflicting measurements of whether the region is occupied or free (for instance due to moving obstacles entering and leaving the region). The mass from the uncertainty set $Z$ will then move to the hypotheses $\{z_1\}$ and $\{z_2\}$, respectively. Hence, the belief will transition from lack of information to conflicting information. In the classic Bayesian counterpart scenario, we would have a uniform prior before a measurement is received, and we would approach the same probability mass distribution when equally many occupied and free sensor measurements are received. As illustrated by this example, evidential theory is able to distinguish lack of information from conflicting information.

### A.3 Evidence Fusion

Two independent sources of evidence represented by belief masses can be combined through Dempster's rule to generate a fused mass function as follows [17],

$$(m_1 \oplus m_2)(A) = \frac{1}{1-\kappa} \sum_{B \cap C = A} m_1(B)m_2(C),$$

$$\forall A \subseteq Z, A \neq \emptyset \text{ and } (m_1 \oplus m_2)(\emptyset) = 0 \tag{14}$$

where $\kappa = \sum_{B \cap C = \emptyset} m_1(B)m_2(C)$ is the degree of conflict between two belief mass functions. Two belief mass functions can be combined through Dempster's rule only if for at least one pair $A \subseteq Z$ and $B \subseteq Z, m(A) \neq 0, m(B) \neq 0$, and $A \cap B \neq \emptyset$ [18]. Dempster's rule reduces to Bayes' rule in the special case of the combination of a vacuous mass function and a mass function with non-zero elements only over singleton sets [18].

## B  DST-Softmax Equivalence Satisfaction

The constraint required to ensure that the DST combination is equivalent to the softmax transformation is: $\sum_{j=1}^{J} \alpha_{jk} = \hat{\beta}_{0k} + c_0$ for some constant $c_0$. Computing, we have:

$$\sum_{j=1}^{J} \alpha_{jk} = \sum_{j=1}^{J} \left[ \frac{1}{J} \left( \beta_{0k} + \sum_{j=1}^{J} \beta_{jk}\phi_j(x_i) \right) - \beta_{jk}\phi_j(x_i) \right] \tag{15}$$

$$= \sum_{j=1}^{J} \frac{1}{J} \left( \beta_{0k} + \sum_{j=1}^{J} \beta_{jk}\phi_j(x_i) \right) - \sum_{j=1}^{J} \beta_{jk}\phi_j(x_i) \tag{16}$$

$$= \frac{1}{J} \left( \beta_{0k} + \sum_{j=1}^{J} \beta_{jk}\phi_j(x_i) \right) \sum_{j=1}^{J} 1 - \sum_{j=1}^{J} \beta_{jk}\phi_j(x_i) \tag{17}$$

$$= \frac{1}{J} \left( \beta_{0k} + \sum_{j=1}^{J} \beta_{jk}\phi_j(x_i) \right) J - \sum_{j=1}^{J} \beta_{jk}\phi_j(x_i) \tag{18}$$

$$= \left( \beta_{0k} + \sum_{j=1}^{J} \beta_{jk}\phi_j(x_i) \right) - \sum_{j=1}^{J} \beta_{jk}\phi_j(x_i) \tag{19}$$

$$= \beta_{0k} \tag{20}$$

$$= \hat{\beta}_{0k} - \frac{1}{K} \sum_{l=1}^{K} \hat{\beta}_{0\ell}. \tag{21}$$

The last line follows from the result in Eq. (8). Therefore, we have shown that the new $\alpha_{jk}$ parameters meet the required constraint for DST-softmax equivalence as posed by Denoeux [19].

## C  MNIST, Fashion MNIST, and NotMNIST Image Generation Experimental Details

We chose the following class reassignment scheme for Fashion MNIST: *tops* and *bottoms/accessories*. For the MNIST and Fashion MNIST experiments, we use a hidden unit dimensionality of 30 for $p(z \mid y)$ and 256 for $p(z \mid x, y)$ and $p(x' \mid z)$ within the architecture in Fig. 1. We chose the 256 dimension following the example from: `https://github.com/timbmg/VAE-CVAE-MNIST`. We use the ReLU nonlinearity with stochastic gradient descent and Adam [42] optimizers and a learning rate of $0.001$ for the MNIST [25] and Fashion MNIST [27] datasets respectively. We train for 20 epochs with a batch size of $64$.

To investigate a similar image generation task on a more complicated dataset, we consider the NotMNIST benchmark, which requires a more expressive architecture. We consider the task of

generating letters with and without a horizontal bar in the center. The convolutional architecture for the NotMNIST experiments is depicted in Fig. 5. Similar to the network for (Fashion) MNIST, during training, the encoder consists of two network blocks. One fully-connected block takes as input the query label $y$, and outputs a softmax probability distribution that parameterizes the prior distribution $p(z \mid y)$, where $z$ is a discrete latent variable that can take on $K$ values. The second block takes as input the stacked feature vector $x$ and query label $y$, and outputs the softmax distribution for the posterior $q(z \mid x, y)$ after a series of convolutional layers. The $z$ value is sampled from the posterior distribution $q$ during training and the prior distribution $p$ at test time. It is then passed through the decoder convolutional network block to predict the image output $x'$. As before, the Gumbel-Softmax distribution is used to backpropagate loss gradients through the discrete latent space [7, 29]. We use the ReLU nonlinearity with the stochastic gradient descent optimizer and a learning rate of $0.00001$. We train for $1000$ epochs with a batch size of $256$.

For the MNIST, FashionMNIST, and NotMNIST experiments, the standard conditional evidence lower bound (ELBO) was maximized to train the model [16]:

$$\mathcal{L}(x, y; \theta) = \mathbb{E}[\log(p_D(x \mid z; \theta))] - \text{KL}[q(z \mid x, y; \theta) \mid\mid p(z \mid y; \theta)], \qquad (22)$$

where $p_D$ is the distribution output by the decoder and $\theta$ are the network parameters.

Figure 5: The CVAE architecture used for NotMNIST image generation. The last layer in each encoder network block is the softmax layer. At test time, $p(z \mid y)$ is used to sample the latent space; thus, only the input query $y$ serves as input to the encoder.

# D  MNIST CVAE Test-Time Performance Comparison

Fig. 6 shows a comparison of the generated images sampled from the softmax discrete latent distribution versus our proposed filtered distribution for the *even* input query. The softmax distribution often samples visually incorrect latent classes given the *even* input query. Our method improves the test time sampling performance of the CVAE by pruning the majority of the erroneous latent classes, while keeping the correct ones. When considering 25 samples for the *even* input, softmax produces eight incorrect samples, while our method produces only three (the incorrect 9 image).

Figure 6: Samples from the CVAE at test time for the *even* input using the softmax distribution and our proposed distribution. Our distribution results in samples that are more accurate with the exception of the sampled 9 digit. The filtered *even* distribution no longer includes the incorrect latent classes for the 1 and 3 digits.

Figure 7: Our proposed distribution (green) extracts more accurate information across fewer training examples on the MNIST dataset as compared to the softmax (blue) and sparsemax (orange) baselines. Lower is better for the distance metrics. The results are over 25 different random seeds.

# E   Reduced Data Performance for MNIST

We investigate whether evidential latent space sparsification is able to extract more accurate information than softmax and sparsemax when the learned model is hindered by a smaller training set. Fig. 7 summarizes the performance of the proposed filtered distribution for a network trained on a reduced MNIST dataset. We use $0.1, 0.5, 0.75$, and $1.0$ fractions of the dataset for training, maintaining the class balance and evaluating after $20$ training epochs. The filtered distribution largely outperforms softmax outside of standard error across both metrics on the MNIST dataset. Our method also outperforms the sparsemax baseline when the latter is hindered by its generation of false negative latent classes (e.g., for the *odd* input query).

# F   Results on Fashion MNIST

## F.1   Qualitatitive Results

We investigate the qualitative performance of our proposed methodology on the Fashion MNIST dataset. The network learns a more accurate softmax distribution than that for the MNIST dataset as shown in Fig. 8. The more effective softmax distribution learned for the Fashion MNIST dataset than that for MNIST is likely due to the more distinct features across the dataset's categories. Nevertheless, our filtered distribution still provides further improvement for the latent class distribution by filtering out incorrect probability masses completely. We note that the filtered distribution makes two mistakes in keeping the first latent class (a boot) for the *tops* input query and keeping the sixth latent class (a shirt) for the *bottoms* input query. Nevertheless, we emphasize that these are false positives. As with MNIST, sparsemax results in aggressive filtering that removes valid latent classes for each query, such as the dress from the *tops* category, and a boot as well as two purses from the accessories

Figure 8: Our proposed filtered distribution (green) is compared to the softmax (blue) and sparsemax (orange) distributions on the Fashion MNIST dataset. The horizontal axis depicts decoded latent classes. Our method reduces the size of the relevant latent sample space without removing valid latent classes.

Figure 9: Our filtered distribution (green) outperforms the softmax (blue) and sparsemax (orange) baselines across training iterations on the Fashion MNIST dataset. Lower is better for distance metrics. The results are over 25 different random seeds.

category. Qualitatively, our filtered distribution outperforms the baselines on the Fashion MNIST dataset, providing a sparser, more accurate distribution than softmax, while avoiding false negatives, which would be undesirable for safety critical applications.

## F.2 Training Evolution Results

Fig. 9 shows that when the underlying network learns the latent space successfully, as is the case for Fashion MNIST data, our filtered distribution performs no worse (and even slightly better) than the original softmax distribution. The sparsemax distribution once again filters out valid latent classes from both binary queries, resulting in poor performance across our metrics. Thus, for the Fashion MNIST benchmark dataset, the proposed latent class distribution provides a more robust representation, retrieving richer information from the learned neural network weights with fewer training iterations.

## F.3 Reduced Data Performance for Fashion MNIST

Fig. 10 summarizes the performance of the filtered distribution on a network trained on a reduced Fashion MNIST dataset. Due to the more effectively learned encoder weights, our filtered distribution maintains the performance of the original softmax distribution. Our proposed distribution significantly outperforms the sparsemax baseline due to the aggressive sparsemax filtering that results in false negatives. Sparsemax continues to select a subset of more likely encodings, at the cost of removing valid latent classes, making it undesirable for applications where false negatives are safety critical. We note that the difference between the Wasserstein and Bhattacharyya metrics in Fig. 10 are due to the latter favoring sparse distributions by definition.

Figure 10: Filtered distribution performance across fewer training samples on the Fashion MNIST dataset. Our filtered distribution (green) demonstrates more robust performance than the softmax (blue) and sparsemax (orange) on our metrics. The results are over 25 different random seeds.

Figure 11: Our proposed filtered distribution (green) is compared to the softmax (blue) and sparsemax (orange) distributions on the NotMNIST dataset. The horizontal axis depicts decoded latent classes. Our method reduces the size of the relevant latent sample space without removing valid latent classes.

Figure 12: Our filtered distribution (green) outperforms sparsemax (orange) and softmax (blue) over training iterations when the network weights have been trained sufficiently on the NotMNIST dataset. Lower is better for distance metrics. The results are over 5 different random seeds.

## G  NotMNIST Image Generation Results

### G.1  Qualitative Performance

Fig. 11 shows a comparison of our proposed method, the original softmax distribution, and sparsemax for the *with middle bar* and *no middle bar* letter queries on the NotMNIST dataset. Once again, although the CVAE architecture proposed in Fig. 5 successfully learns a multimodal latent encoding, the learned softmax probability distribution has non-negligible probability masses associated with the incorrect latent classes $z_k$ for each query class $y$.

We observe that our filtered distribution selects a plausible set of correct latent classes given an input query as shown in Fig. 11. Since NotMNIST is a more difficult dataset, the decoded latent classes are, at times, an unrecognizable combination of features which scores high for both input queries. Outside of these cases, we successfully sparsify the latent space analytically for each query class without knowledge of or comparison to the other query. As before, sparsemax continues to generate excessive false negatives, for instance, filtering out the 'I' for the *without middle bar* query. Thus, our more conservative, yet effective, filtration is a compelling latent space reduction technique.

### G.2  Quantitative Performance

**Training Evolution**  Fig. 12 shows the performance of our filtered distribution on the NotMNIST dataset at different training iterations. For the *with middle bar* input query, we continue to demonstrate the robustness of our filtered distribution to fewer training iterations. Our method extracts more accurate encoding information from the neural network earlier in the training process than the softmax and sparsemax baselines. However, we note that both our filtered distribution and sparsemax

underperform the softmax baseline at the beginning of training for the *without middle bar* query. Since the NotMNIST dataset is substantially bigger and more difficult than (Fashion) MNIST, the neural network weights governing the latent distribution can take longer to become meaningful. Hence, we observed that for these early iterations, it is not possible for a method with an implicit threshold, such as our filtration or sparsemax, to accurately filter the distribution. As in other experiments, our filtered distribution can only perform as well as the learned network. However, our distribution continues to consistently outperform sparsemax due to the latter's aggressive filtration which incorrectly filters out potentially valid latent classes.

# H  VQ-VAE Experimental Details

We train the VQ-VAE [2] network on *mini*ImageNet [26] data randomly cropped to $128 \times 128$ and normalized to $[-1, 1]$. The *mini*ImageNet dataset consists of $38,400$ examples in the training set, $9,600$ in the validation set, and $12,000$ in the test set. *mini*ImageNet was designed for few-shot learning tasks, thus the classes do not overlap between the dataset splits. There are 64 classes in the training set. We use *mini*ImageNet to test the algorithm due to its more computationally feasible size for training on a single NVIDIA GeForce GTX 1070 GPU. We train the VQ-VAE with the default parameters from: `https://github.com/ritheshkumar95/pytorch-vqvae`. We use a batch size of 128 for 100 epochs, $K = 512$ for the number of classes for each of the $32 \times 32$ latent variables, a hidden size of 256, and a $\beta$ of one. The network was trained with the Adam optimizer and a starting learning rate of $2 \times 10^{-4}$. We use the best model according to the validation loss. To sanity check that the VQ-VAE latent space reasonably captures the data, we demonstrate example input and output images from the *mini*ImageNet test set in Fig. 13. Since *mini*ImageNet is meant for one-shot learning, the classes in the training set do not match those in the validation and test sets. Thus, the data distribution in the test set is different than that of the training set. The trained VQ-VAE is able to reasonably reconstruct these out-of-distribution images. We then train the PixelCNN [31] prior over the latent space with 20 layers, hidden dimension of 128, a batch size of 32 for 100 epochs. The network was trained with the Adam optimizer and a starting learning rate of $3 \times 10^{-4}$.

We generate a new dataset by sampling from the trained prior, and decoding the images using the VQ-VAE decoder. We sample 25 latent encodings from the prior for each of the 64 *mini*ImageNet training classes to build the dataset. We perform the sampling using the original softmax, sparsemax, and our proposed latent distributions. We extract equivalent linear layer weights and biases for the last 1D convolutional layer in PixelCNN to pass as input to our proposed distribution and sparsemax. Examples of the sampled images are shown below in Fig. 14. Our proposed sparse latent distribution visually maintains the same performance as the softmax distribution. Images sampled using the sparsemax distribution are not depicted as they degenerate to single color blocks due to sparsemax severely collapsing the distributional multimodality in the PixelCNN prior.

We then train a Wide Residual Network (WRN) [32] for classification on *mini*ImageNet. We use the PyTorch implementation for WRN found here: `https://pytorch.org/docs/stable/torchvision/models.html` and the training protocol proposed here: `https://github.com/huyvnphan/PyTorch_CIFAR10`. We train WRN on a subset of the *mini*ImageNet training set, leaving $5\%$ out for validation. WRN is trained for 100 epochs with a batch size of 128. The optimizer is stochastic gradient descent with a learning rate of $1 \times 10^{-2}$. The inference performance of the WRN classifier on the datasets generated with the softmax, sparsemax, and our proposed latent distributions are compared, demonstrating that our distribution, unlike sparsemax, is able to maintain the performance of softmax, while significantly reducing the size of the latent sample space.

(a) Original                    (b) Reconstructed by VQ-VAE

Figure 13: Images are reconstructed using VQ-VAE from the test set of *mini*ImageNet.

|                    |                    |
|:------------------:|:------------------:|
| (a) Softmax        | (b) Ours           |

Figure 14: Images are generated for the queries (top to bottom): "cliff", "jellyfish", "orange", "rock beauty", and "yawl" using the original softmax and our proposed latent distributions. Both sets of sampled images are of similar quality despite our distribution considering a much smaller latent sample space. Images sampled using sparsemax are not depicted as they degenerate to solid color blocks due to sparsemax collapsing the multimodality in the autoregressive latent space distribution.

## I  Behavior Prediction Experimental Details

Fig. 15 illustrates the architecture of the Trajectron++ network with internal layer sizes for reference, as depicted in Salzmann et al. [4]. The model was trained for 2000 iterations with a batch size of 256 and an initial learning rate of $1 \times 10^{-3}$ which was exponentially annealed down to $1 \times 10^{-5}$ with a decay rate of $0.9999$. The model was trained to predict 12 timesteps ($4.8\,\text{s}$) into the future from 8 timesteps ($3.2\,\text{s}$) of history. The loss function $\beta$ weight was varied following a sigmoid function from 0 to 2.5 with the middle value achieved at 400 iterations. The model's latent variables $\mathbf{z}$ are one-hot categorical latent variables which are approximated with a Gumbel-Softmax distribution, enabling backpropagation through samples with the reparameterization trick [7]. The Gumbel-Softmax distribution's temperature $\tau$ was exponentially annealed from $2.0$ to $0.05$ with a decay rate of $0.997$. The decoder outputs Gaussian Mixture Model (GMM) means and covariances for each prediction timestep, where each GMM has 16 components. Our experiments were run on 50 randomly-sampled scenes from the ETH test dataset, within which there were 203 agent trajectories.

Figure 15: Trajectron++ architecture with layer dimensions indicated.