[Reviews · NeurIPS 2020]

Review 1

Summary and Contributions: This paper introduces a technique for sparsifying the discrete latent space of a CVAE. The method is grounded in evidential theory and is applied post-training: a mass function equivalent to the softmax transformation is constructed such that the mass over singleton sets is sparse. Singleton sets (i.e. latent classes) with mass 0 are then filtered out and the distribution is normalized. They demonstrate that their method effectively sparsifies the latent space while maintaining more modalities than other sparsification techniques in the domains of image generations and trajectory prediction. --- Post-rebuttal update --- The authors adequately addressed my concerns in their rebuttal, and I've increased my score to a 7. I think a discussion about tuning the level of sparsity is important (also pointed out by another reviewer) and should be included in the main text. Some results about static thresholds for MNIST would also be great to have in the appendix.

Strengths: 1) Strong empirical results - the method seems to be on-par with softmax while also sparsifying the latent space, whereas sparsemax clearly drops modes and performs worse. Multiple domains are evaluated. I appreciate the proof-of-concept experiments with MNIST to build intuition. 2) The paper is well motivated and very well written. I’m not too familiar with evidential theory, but I was able to follow along in section 2. This is an interesting perspective of classification / supervised learning that I haven’t encountered previously.

Weaknesses: 1) The biggest difference between this and sprasemax is that this is applied post hoc whereas sparsemax is differentiable and applied during training. Shouldn’t a fair comparison also consider other sparsification methods that can be applied post hoc? For instance in Figure 2 for the MNIST experiment, it looks like the method selects the top k largest probability masses. Could you also consider a naive baseline method where the probabilities below a certain threshold are set to 0 and redistributed? Of course you will need additional tuning of the threshold and the method is not as theoretically grounded, but I’m just curious how this naive method would perform. 2) There are also other sparsification techniques that are differentiable and can be applied during training, such as the ones in [1]. The paper can be strengthened by comparing with more techniques (either in discussion or additional experiments, or both). [1] Laha et al. On Controllable Sparse Alternatives to Softmax.

Correctness: Yes

Clarity: Yes

Relation to Prior Work: Missing reference to [1].

Reproducibility: Yes

Additional Feedback: - In Eq. 8, what’s \hat{\beta_{jk}} ? - Is there a typo in Eq. 21 in the appendix? Should the second term be the mean of all bias parameters? - A brief summary of sparsemax would help (maybe in the appendix)


Review 2

Summary and Contributions: The paper introduces a methodology for reducing the dimensionality of the discrete latent space in variational auto-encoders. The approach is based on the principles of evidential theory, that authors use to filter out the latent classes that do not capture evidence of the data.

Strengths: In general the paper is well-written, and demonstrates a high-consistency and robustness in the flow of technical explanations. I particularly appreciate the motivation and introduction of the work, something that was not easy to do that well in this case. Moreover, the authors did a great job in the synthesis of evidential theory in the beginning of section 2. The idea of using the evidence formulation of Dempster (2008) for a posterior decision-making of what latent classes can be removed is definitely interesting and introduces a new brach of work in the design of discrete latent spaces. This last point is the one that I remark as the most important and novel in the work. I did not find or remember a similar approach on this, and neither the paper is super-conditioned to some other previous work. Last but not least, authors dedicated 4 pages to experiments, evaluation, figures and discussion, something that is valuable and proves the spirit of the authors for demonstrating the performance of the solution.

Weaknesses: I find three points of weakness that decrease the potential impact of the work: i) References are too focused on “application” papers and evidential theory, while authors want to present a new methodology for reducing the discrete latent space dimensionality in auto-encoders. Why do I say this? Well, if authors include more references or comments about theoretical papers of VAEs, this work could be better contrasted with other similar works, and will potentially facilitate its disclosure.. ii) Apart from the references, authors fail on the fact of not including a short paragraph or subsection about the CVAE with a few details to refresh the ideas and having a work that is totally self-contained. They could have sacrificed half-page of experiments to described the conditional auto-encoder better. What a pity! iii) I appreciate the effort dedicated to the experiments, however, I have a few questions that I would like to see answered: the error with the number 9 as “even” in Figure 2 is partially explained, ok. could the approach amplify errors or mistakes in the discrete latent space? that is, the good part of uncertainty is that it is never too certain or the opposite. So, if the number 9 was badly compressed in the latent space, and then so many other dimensions removed, after re-normalising, the number 9 gets importance? is that what is happening? The other question is about Table 1 and the accuracy performance under the 50% in classification, pretty bad, right? why is this? how could be improved?

Correctness: I do not detect any mistake or error in the paper, but I would like to see the latent classes z_k and its domain explained a bit better. I know authors refer very well to the softmax transformation, but, the reader does not know if the z_k are one-hot-encodings, natural numbers, real values, etc. This fits in the paragraph of L58 or later in the description of L68.

Clarity: The paper is clearly written and amenable to be read. The last sentence of the conclusions (L309) could be reformulated or omitted. “We leave the investigation of evidential latent space … to future work…” sounds a bit unmotivated. Why not “In future work it would be interesting to explore the dimension or the characterization of the ev. latent space”, something like.

Relation to Prior Work: My concerns about the prior work related to theoretical or more “pure” VAE papers where described in the previous sections of the review.

Reproducibility: Yes

Additional Feedback: Minors: L58: lack of information from conflicting information? information from information? Paragraph L58-L65 a bit repetitive wrt the end of the intro. Could it be elaborated a bit more? L95: what is “j”? what is indexing? L93—L102: more dense part of the paper, some extra information in the appendix perhaps? for helping non-familiar readers? L146: “these real-world (…) require high degrees of distributional multimodality”, ok, I trust the statement, but why? elaborate a bit please. Figure 2: your columns in the histogram should appear in a darker color when printing in grey-scale. otherwise there is no difference and I had to go to the original colored pdf. ###### [Post-rebuttal update] ######## I appreciate the time dedicated by the authors to answer the 3 points of weakness that I commented in the initial review. Looking again to the paper, I find the idea definitely interesting and the advances on the design of discrete latent spaces will have an impact in the future. In general, I think that it is a good paper and my concerns have been addressed in the response, so I revised my score upwards.


Review 3

Summary and Contributions: This paper presents a method to sparsify the latent representation of a Conditional Variational Autoencoder. Experiments show the presented method outperforms sparsemax and it is competitive to softmax while sparsifying the latent space.

Strengths: The paper presents a nice theoretical grounding of the method. The technical experiments appear to be correct and they show the presented method outperforms the current sparsemax method.

Weaknesses: The presented method outperforms sparsemax in terms of accuracy, but sparsemax is reducing the latent space by a larger %. In experiment 3.1 sparsemax reduces it by 99% while the presented method reduces it by 89%. Is it possible to adjust what % of the embedding space we want to reduce? In that case I would encourage the authors to present an analysis of how the performance of their method drops as the reduction of the embedding space increases. And then analyze which method provides a better trade off of accuracy vs reduction. In case it is not possible to tune the reduced % of the embedding space and it is just a consequence of the method itself, I would encourage the authors to state that in the paper. If the authors give a convincing explanation to this I am eager to increase the rating of the paper.

Correctness: The methodology appears to be correct.

Clarity: The paper is well written and easy to follow. The evidential theory section was harder to follow to me, I was not familiar with it.

Relation to Prior Work: The work is properly contextualized.

Reproducibility: Yes

Additional Feedback:

[Author Response · NeurIPS 2020]

We thank the reviewers for their thoughtful feedback, and address the main questions raised in turn.

**R1:** 1) We clarify that although sparsemax can be applied during training, it can also be applied post hoc, just like
vanilla softmax. Since we present a post hoc sparsification method, we apply sparsemax post hoc in our experiments
for fair comparison. Both our method and sparsemax compute implicit thresholds for the distribution. However, both
methods (1) do not need to be tuned for each network or dataset, and (2) automatically adapt to individual input features,
unlike a static threshold. We consider a static threshold baseline in our MNIST experiment in Fig. 2 for intuition. We
select a static threshold of $0.14$ using cross-validation with the NLL metric. However, this static threshold caused even
more false negatives than sparsemax. By visual inspection, it is impossible to choose a single static threshold that
would outperform our method in balancing sparsity and multimodality in Fig. 2 (either additional false negatives or
false positives would result). While we could tune a static threshold for each individual input query for the MNIST task
(even/odd), this would be intractable for a continuous trajectory input query as in the behavior prediction task.

2) We will add discussion of the work by Laha et al. to our paper. They augment the sparsemax technique to control
sparsity, presenting sparsegen-lin (sparsemax with regularization) and sparsegen-hg (sparsemax with a scaled input).
Both of these methods require hyperparameter tuning which may be computationally prohibitive on larger networks.
We applied these methods to the MNIST experiment in Fig. 2, and found that they either pruned the same latent classes
as sparsemax or, with certain parameter choices, produced even sparser distributions with more false negatives. Our
method better balances multimodality and sparsity while adapting to different feature inputs *without parameter tuning*;
indeed, such an "auto-tuning" property is a key advantage of our method.

In Eq. 8, $\hat{\beta}_{jk}$ are the linear layer weights learned by the neural network a priori, akin to the definition of the bias
parameters (line 106). We will update our paper to include this definition, a brief summary of sparsemax, and a
correction to the typo in Appendix B, Eq. 21.

**R2:** i) In addition to application-based papers and evidential theory in deep learning, our literature review covers
(1) theoretical works on training VAEs with discrete latent spaces and (2) existing alternatives to softmax (see Sec. 4).
There are indirectly related works in low-dimensional encodings for VAEs that focus on regularization [1] and enforcing
structure in the latent space [2] during training, but they do not sparsify the latent space post hoc. We will include more
VAE references in our updated literature review, and welcome further suggestions.
[1] Mathieu et al. Disentangling Disentanglement in Variational Autoencoders. In ICML, 2019.
[2] Kosiorek et al. Sequential attend, infer, repeat: Generative modelling of moving objects. In NeurIPS, 2018.

ii) Currently, we introduce the CVAE in lines 33–38 and provide details of the CVAE architectures we consider in lines
160–179, Fig. 1, and in Appendices C, H, I. We will update our paper to further describe CVAEs (e.g., the ELBO loss).

iii) We address R2's question: "could the approach amplify errors or mistakes in the discrete latent space?" as
follows. By construction, both our method and sparsemax amplify the likelihood of the latent classes selected by each
technique. However, our method strikes a better balance than sparsemax between sparsity and multimodality of the latent
distribution. As demonstrated in our experiments, specifically in the MNIST experiment referred to by R2, our method
improves the test time sampling performance of the CVAE by pruning the majority of the erroneous latent classes,
while keeping the correct ones (see Appendix D, Fig. 6). When considering 25 samples for the "even" input, softmax
produces eight incorrect samples, while our method produces only three (the incorrect 9 image). Conversely, sparsemax
severely collapses the uncertainty in the latent distribution, removing correct latent classes from the distribution.

Regarding R2's question about the accuracy in Table 1, this performance is due to the quality of the VQVAE images
generated using the PixelCNN prior network. With additional computational resources and the full ImageNet dataset
(as in the VQVAE paper), this performance could be further optimized. However, we were still able to learn sufficiently
good quality encodings using the VQVAE architecture and generate reasonable images (see Appendix H) to demonstrate
the relative performance of softmax, sparsemax, and our proposed method on the latent distribution of the VQVAE.

We will include that $z_k$ are one-hot encodings in line 68. In line 95, $j$ is the index into the feature vector (defined in
lines 93–95). Regarding R2's question on why modeling real-world tasks requires highly multimodal distributions,
consider the example in lines 31–32. If a prediction model receives a trajectory of a person walking straight, the model
should predict multiple plausible paths (e.g., continuing straight or turning left or right), constituting multimodal output.

**R3:** In both our method and sparsemax, it is not possible to manually tune the level of sparsity achieved in the latent
space. Rather, by design, both methods compute an implicit threshold for each input query. Our method automatically
balances the objectives of sparsity and multimodality by keeping only the latent classes that receive direct evidence
from the network's features and weights, as described in Secs. 2.3–2.4. Although sparsemax reduces the latent space
by a larger percentage than our technique, this negatively impacts its performance; our experiments demonstrate that
sparsemax collapses the multimodality in the latent space distribution, resulting in undesirable pruning of correct latent
classes. We convey these ideas in lines 199–204, but we will make them clearer in an updated version of the paper.

[Meta-Review · NeurIPS 2020]

All the reviewers were relatively positive about this paper but there were some concerns, mainly with regards to details of the experimental comparison and related work. These have been clarified during the rebuttal and the reviewers were happy to recommend acceptance.